# When Design Fiction Meets Geospatial Sciences to Create a More Inclusive Smart City

**Andrée-Anne Blacutt** [1] and **Stéphane Roche** [2,*]

1 School of Design, Laval University, Edifice La Fabrique, 295 Boulevard Charest E bureau 090, Québec, QC G1K 3G8, Canada; andree-anne.blacutt-grenier.1@ulaval.ca
2 Research Center for Geospatial Data and Intelligence, Laval University, Pavillon Louis-Jacques Casault, 1055 Avenue du Séminaire, Bureau 2322, Québec, QC G1V 0A6, Canada
* Correspondence: stephane.roche@scg.ulaval.ca

**Abstract:** Smart cities are especially suited for improving urban inclusion by combining digital transition and social innovation. To be smart, a city has to provide every citizen with urban spaces, public services, and common goods that are effectively affordable, whatever the citizen's gender, culture, origin, race, or impairment. Based on two design workshops, the "Vibropod" and the "Pointe-aux-Lièvres", this paper aims at highlighting the contributions of design fiction to the improvement of the spatial capability of hearing impaired people. This research draws its originality from both its conceptual framework, built on an interdisciplinary and intersectoral composition of arts and sciences, and its operational approach, based on the use of the DeafSpace markers and the TRIZ theory (Russian acronym for Inventive Problem Solving Theory) principles. The two design fiction workshops demonstrate that considering the singularity of the human being as an actual acoustic material constitutes an innovative opportunity to improve the role of universal design in a smart city project. By reversing the classic posture, and defining disability by looking at characteristics of the environment rather than as limits of the people themselves (their bodies or their senses), this research proposes an innovative way of addressing smart city inclusivity issues. This paper shows how increasing spatial enablement and having better control of spatial skills can offer deaf people new skills to improve the use of technology in support of urban mobility, as well as give them tools for feeling safer in urban environments.

**Keywords:** deafness; deafscape; design fiction; smart city; spatial skills; social inclusion; TRIZ theory; urban mobility

## 1. Introduction

Smart cities are often presented as the main solution for urban areas to address the global urbanization issues they are currently facing (demography, climate change, social tensions, etc.) [1], as well as a way of modernizing city operations (urban engineering) [2], and governance (openness and transparency) [3]. The smart city's triple and quadruple helix [4] models have, thus, largely become a reference. The vast majority of works show that the principal smart dimension of a smart city is based on a better integration of the digital transition into complex urban dynamics (economic, socio-political, and environmental) [5], as well as on a fundamental renewal of citizen engagement modalities [6].

One other way forward is to define smart cities in terms of stages or building blocks. Cohen [7] identifies three simultaneous generations of Smart Cities. Smart City 1.0—"Technology Driven"—refers to city projects that were "pulled" by technology. Large Information and Communication Technology (ICT) companies, such as IBM or CISCO, have initiated large-scale projects to demonstrate that their

technological solutions and services answer contemporary issues in cities. Cohen [7] considers the second generation to be Smart City 2.0—"Technology Enabled, City-Led". Smart City 2.0 comes forth when cities realize that, to take advantages of a digital transition and global location age, smartness needs to be considered as a form of governance transformation and social innovation [8]. Barcelona is one of those cities, followed by other European cities, such as Paris, Nantes, Copenhagen, and Helsinki. Cities realized they could improve their governance to ensure metropolitan prosperity by both managing community and citizen engagement through digital innovations, as well as articulating competition and cooperation [9]. More recently, a third generation, a Smart City 3.0 [7] has emerged, putting citizens at the heart of urban innovation schemes. Called "Citizen Co-Creation", Smart City 3.0 is finally rooted in a larger quest for inclusion, equity, and justice.

More specifically, the need and the relevance of combining digital transformation, on the one hand, and social innovation, on the other hand, is mainly justified by the increasingly expressed desire to turn urban spaces and places into more ethical, fair, and inclusive environments [10]. To be smart, a city must be inclusive [11]; however, urban inclusion is a complex issue that should be addressed from multiple perspectives (social dynamics, gender, racial issues, disability, etc.), using various approaches, methods, and techniques [12]. Urban inclusion also has different components (economics, health, technology, etc.), and is still very sensitive to scales (social, spatial, and temporal). Most of the research that has been done so far with regards to urban inclusion were dealing with spatial accessibility and mobility [13]. However, the dynamics surrounding smart city projects, practices, and researches, bring new inclusion issues to the forefront (not only spatial ones), and new ways of embracing those issues [11]. Indeed, the dematerialization of municipal public services, interactions, and communication, as well as the growing need to use digital interfaces, and mobile and smart technologies, have generated a new form of urban exclusions (digital divides, digital literacy, data deserts, online services accessibility, etc.) [14].

In this context, universal design has been mobilized [15] to support the conception, development, and implementation of more inclusive assistive technologies, and to make urban spaces and functions more affordable for people with different forms of disabilities or handicaps [16]. More precisely, together with the Disability Creation Process (DCP) model, human-centric and User Experience (UX) design approaches have emerged as an interesting and useful way for some researchers to begin explorations under the smart city umbrella [16]. Using the DCP model aims to define disability with regards to the characteristics of the environment rather than as the limits of the people themselves (their bodies or their senses) [17], which is actually a very innovative and relevant way of facing urban exclusion challenges.

This article focuses on issues and challenges of deafness in urban environments and the use of spatially enabled design fiction methods as a way to make smart cities more Deafscape compliant [18]. Our approach is, therefore, fully in line with the Smart City 3.0 model [7]. Hearing impaired people, even those with cochlear implants, experience new communication challenges (sensorial, aesthetic, physical) linked the use of the prosthesis [19]. In particular, the use of a cochlear implant induces a sort of illusion of normal hearing that could alter the perception of the condition, skills, and needs of deaf users, and of the role of the implant itself. Due to those challenges, urban mobility remains a risky activity for implanted hearing impaired people. They need urban sound information (characteristics of urban soundscapes) to help them qualitatively understand urban spaces. However, the vast majority of the research and development concerning deafness (in particular about cochlear implants) mainly focuses on speech aid, but little on sound spatialization.

The main aim of the work presented here is to highlight the contribution of design fiction in the improvement of the spatial capability of deaf paired pedestrians. Two design fiction workshops were organized, and showed to what extent increasing spatial enablement, and better control of spatial skills, offer hearing impaired users of cochlear implants, regarding an increased capacity to improve the use of technology in support of urban mobility.

Section 2 presents the main components of the conceptual and theoretical framework this research is based on: first, inclusive smart city researches are summarized, in order to help present more specific

works, such as the ones conducted on deafness, Deaf Gain, and DeafSpace. The DeafSpace concept is very relevant insofar as it makes the link with the smart city. Finally, relevant questions regarding spatial-enablement and spatial skills are presented. In Section 3, the two design fiction workshops are described (approach, methods, tools, results, etc.). Section 4 focuses on a discussion and provides feedback on experiences in relation to smart city inclusiveness. Some very short conclusive remarks are proposed in Section 5.

## 2. Conceptual Bases

### 2.1. Inclusive Smart City

Smart city concepts and technologies appear as the most effective available solution to tackle urban challenges. Yet the technological approach of a smart city undoubtedly raises as many issues as answers [10,20]. In particular, the technological solutionism on which smart city projects are still too often based, raises issues of privacy, security, and ethics, as well as of social and spatial inclusion.

Smart city technologies, such as sensors, IoT facilities, data analysis tools, personal apps, etc., are usually designed and developed for an average, typical user [21], not for users with special needs or disabilities. Research has already clearly demonstrated to what extent those technologies fail to support both social and gender or, even more, disability inclusions [22]. Those technologies especially lack adaptability and accessibility [23]. This is particularly the case for geospatial technologies (navigation or wayfinding tools for instance) [24]. Geographical (urban) accessibility or safety issues are quite well documented in the literature [25]. Data desert and data poverty [26], digital divides [27], and socio-economic inequalities with regards to technology and data accessibility [28,29], have also been studied to a significant extent.

The way to enable disabled people to benefit from the advances of a smart city is certainly to consider their specific needs as a foundation of any project. As mentioned by Lise Wagner [23], "*Just like physical accessibility for roadways and buildings, the cost for digital accessibility is almost minuscule when integrated into the design brief at the very beginning*". In 2016, Smart City for All [30] conducted a survey of experts around the world, showing that the lack of awareness about disability and accessibility in design and innovation typically appears as one of the main impediments to building a more inclusive smart city. Smart City for All is an initiative that aims at defining the state of and requirements for better technological accessibility in smart cities worldwide. Its main focus is to bridge the digital divide for persons with disabilities and older persons in Smart Cities. Smart City for All has developed a toolkit composed of four resources: a guide to implementing technological accessibility standards, a guide to adopting technological accessibility policies, a communications package, and a database of demos and proofs of concept.

Some projects try to provide answers, with multidisciplinary teams involving researchers in engineering, social sciences, medicine, and rehabilitation, created to address these questions in a systemic and holistic manner [24]. The MobiliSIG [28] project developed within Center for Interdisciplinary Research in Rehabilitation and Social Integration (CIRRIS) at Laval University (Quebec) is a good example of this kind of synergy [31]. The CIRRIS team conducts research activities studying the personal factors (impairments and disabilities) and the environmental factors (barriers and facilitators) that influence the social participation of persons with disabilities. Their research projects are focused on interdisciplinary and intersectoral approaches that allow the study of complex issues through the integration of biomedical and social research. In this context, MobiliSIG aims at designing and developing a multimodal technological solution for mobility assistance aimed at people using a manual wheelchair or a scooter, all in an urban context [32].

The amount of research that aims at improving the smart city inclusiveness is actually increasing. Some of it addresses very technical issues and uses high-tech innovations (augmented reality, artificial intelligence, etc.) for improving the accessibility of people with motor disabilities [25]. Urban navigation and wayfinding specific problems are often addressed by using geospatial sciences and engineering

facilities, as is the case with the MobiliSIG project [28]. The development of more inclusive and ethical smart city platforms is also regularly studied [33,34] in order to make online municipal services disabled-user-friendly, and as a way of using data analysis to better understand the specific urban stakes that citizen with disabilities may face. Research about adapted digital interfaces (mobile or fixed) has also developed quite a lot [29,35,36]. Another very important field of study is the design of urban places and spaces itself. Indeed, as previously mentioned, making a smart city disabled-friendly is not just a question of developing adapted technologies, it is also a way of thinking about urban environments as inclusive ones [37,38]. Very little research has addressed this question from a smart city perspective [39].

However, people with disabilities in general, and deaf people in particular, are most often overlooked when smart urban areas are designed and planned [22,36]. Numerous studies show how much the sensory characteristics of deaf people require that their life and mobility spaces be adapted [40], that the design of their components take into account deafness [18,41], and that the design of assistive technologies modulated interactions between the deaf and their spaces to anchor them in the state of deafness [18,41]. Yet, to be smart, a city has to be ethical and inclusive. A smart city is also (even essentially) a city where citizens (not only a specific and well-off category of the population) can access social innovations and take the benefit of the technical ones in support to their daily urban life [6].

## 2.2. Deafness: Deaf Gain, Deafscape, and Cochlear Implants

In the field of smart city studies, a growing number of works and research projects have been conducted in the domain of disabilities and their interactions with assistive technologies and specific environments [42]. In the case of deafness, more particularly, the research dealing with Deaf Gain and DeafSpace is probably the most singular, and the most in line with the idea of improving the smart city inclusiveness [43]. Deaf Gain is a fairly new term often used by people of the deaf community (as they define themselves [41]) when discussing all of the benefits of being deaf and being involved in the deaf community. Rather than seeing deafness as hearing "loss", many deaf people prefer to see it as a deaf "gain" [41]. By considering the unique cognitive, cultural and creative dimensions of deaf culture, DeafSpace is certainly the most effective means by which deaf culture, in all its various dimensions, can be developed [41]. Experimentations and prototypes have mainly been developed by the teams of the Department of American Sign Language and Deaf Studies of the College of Arts and Sciences at Gallaudet University, where the concepts of Deaf Gain and DeafSpace were developed. The DeafSpace approach brings together deaf culture and universal design in order to rethink the environment (rooms and housing, urban places and spaces, etc.) from a 360-degree learning approach that aims at improving deaf people's skills (peripheral, transparency, reflection, vibration, and shared sensory reach) [18].

The 360-degree learning approach for DeafSpace has four specific objectives for assisting deaf people: (1) improving their capacity of feeling their sensations; (2) identifying better ways to express their experiences; (3) being more confident about getting in touch someone else; and (4) being able to observe their environment as well as the connections between objects and individuals. DeafSpace is an architectural concept tailored to deaf vision and culture in space. Buildings, hallways, stairs, and other spatial arrangements are designed to deaf people's ways of seeing and being in their environments. The first experiment of planning and building a DeafSpace was initiated by Hansel Bauman (architect) in 2005 with the ASL Deaf Studies Department at Gallaudet University, in collaboration with his brother, Dirksen Bauman, who is a professor of Deaf Studies at Gallaudet University. Mobility and proximity, space and proximity, acoustics (vibration), lights (shadow, transparency, color), and sensory reach (refers to the needs of Deaf people to be spatially oriented and visually aware of the activities in their surroundings) are essential components of the environment in DeafSpace [18].

In the context of DeafSpace, 360 degrees means providing accessibility to the five required markers/skills for a better understanding of space for deaf people: peripheral, transparency, reflection,

vibration and shared sensory reach [18]. Together, these five DeafSpace markers (Figure 1) form the components of the 360-degree spatial learning equation:

- Peripheral refers to the use of rhythmic, repetitive, and intuitive visual cues to support the deaf person's peripheral vision;
- Transparency refers to visual connections, openness, and of the degree of the enclosure;
- Reflection refers to the extension of vision, allowing deaf people to see behind them, and to access depth and perspectives;
- Vibration refers to the characteristics of the floor surfaces that allow to feel the presence of others and initiate contact;
- Shared sensory reach refers to the interdependency of deaf individuals when they navigate their environment, as they strongly depend on one another to extend their spatial and orientation skills and spatial reasoning capabilities.

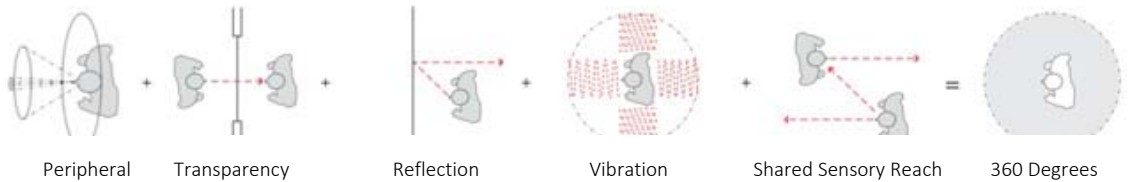

**Figure 1.** Five markers of DeafSpace (source: ©Dangermond Keane Architecture).

On the other hand, it is interesting to note that universal design is being more and more mobilized in order to support the conception, development, and implementation of more inclusive assistive technologies [44], and to make urban spaces more accessible for people with different forms of disabilities [45]. More precisely, together with the Disability Creation Process (DCP) model [46], human-centric and UX design approaches have emerged as interesting and useful ways of thinking about the management of disability [40]. Indeed, the use of the DCP model aims at defining disability in terms of the characteristics of the environment rather than as the limits of the people themselves (their bodies or their senses) [17]. This model aims at explaining the causes and consequences of diseases, trauma, and other effects on people's integrity and development. The DCP is a model that does not put the responsibility of disability on the person [29]. From this model, the understanding and explanation of the disability phenomenon are based on the interaction between three conceptual domains: personal factors (organic system, capability of the person, and identity factor), environmental factors (facilitators vs. obstacles/social vs. physical), and life habits (social participation situation vs. disabling situation/current activities vs. social roles) [17].

The situation of hearing impaired people with a cochlear implant is singular. Indeed, at the beginning, the sound sensations perceived by implanted users may not correspond to the sensations of normal hearing, or those of hearing externally paired. This is the reason why a specific hearing education practiced with an audiologist or a speech therapist (a professional who treats and rehabilitates communication disorders linked to hearing) is almost always necessary for proper cochlear implant integration. The cochlear implant is an electronic device that allows people with severe to profound deafness to have better access to sound. It consists of an internal part inserted under the skin, behind the ear during surgery performed under general anesthesia; and an external part, a voice processor connected to the antenna.

Although the helpful nature of these implants is widely demonstrated, a certain number of technological, cognitive, and social limits remain [47,48]. Moreover, political issues have to be taken into account. Indeed, depending on the country (especially its medical ecosystem), cochlear implantation has been either highly recommended or absolutely boycotted. For instance, while France was one of the pioneering countries for these devices, the development of implants and their deployment were long opposed by specialists [49,50]. Advances in biotechnology and medical engineering have made it

possible to achieve a demonstrated level of effectiveness of cochlear implants, particularly in cases of deep or congenital deafness. However, the most recent research unambiguously demonstrates that without close and sustainable support, without monitoring and active engagement of all the components of the healthcare ecosystem (medical specialists, audiologists, specialized educators, etc., but also caregivers, parents and patients themselves), the success rate decreases drastically, and the risks of indirect negative effects increase considerably, such as psychological disorders or even explant. [19,51,52].

Aligning once again with the Deaf Gain and DeafSpace concepts, the DCP model certainly offers another interesting way for encompassing the necessary engagement of deaf people and their relatives in the overall management of their urban interactions (social and spatial), and the consideration of deafness, not as a disability, but rather as an opportunity for designing smart cities from a broader perspective.

## 2.3. Spatial Skills, Spatiality

Citizen engagement is indeed one of the most important requirements for making cities smarter [17]. The intelligence of a city should be measured by its ability to produce favorable conditions for citizens, organizations, private companies, and other urban operators to be actively involved in socio-spatial innovation [6]. However, even more, a smart city is a city where urban stakeholders have developed an ability to identify what is happening in the city and to react appropriately in a relevant time frame, as part of their spatial capital, something that, today, relies both on spatial enablement (spatial skills and spatial thinking capability) and digital literacy [20,21].

Urban citizens should then be able to develop and mobilize digital spatial skills to manage their spatiality. The elementary skills of human spatiality, i.e. spatial skills, could be defined as the specific capacities of individuals required to accomplish and assume their spatial activities. These cognitive and practical (performative) capacities are complementary and matched [53]. According to Lussault [53], people master their interactions with (and in) urban spaces through the six following skills: (1) the metric skill refers to the ability of people to master their spatial relations of distance (in the context of the pandemic, for example, this competence is useful for understanding the rules of social distancing). (2) The location skill is useful to people not only to choose the best places for their activities, but also to manage interactions with other personal spaces. (3) The course skill is the wayfinding and routing abilities of people—a skill particularly relevant in the context of urban mobility. (4) The crossing skill brings together all the acquired techniques and practices, which allow people to cross (or attempt to cross) airlocks, thresholds, borders, security gates, and all kinds of limits people face in their daily life. (5) The spatial delimitation skill is a double capacity to divide the space into relevant elementary units and to delimit, to set spatial limits between the different discriminated entities (competence of division and delimitation). Last, but not least, (6) the scalar skill refers to the ability of people to distinguish the small from the large and, thus, to apprehend the absolute and relative size of objects, phenomena, and spatial dynamics.

Following this line, a smart city is, first of all, a spatially enabled city. Spatial enablement means that "*location, place and any other spatial information are available to governments, citizens and businesses as a means of organizing their activities and information*" [54]. So, a smart city is a city where urban stakeholders have developed this ability to identify what is happening in the city and to react appropriately in a relevant time frame, as part of their spatial capital, which today relies both on spatial enablement (spatial skills and spatial thinking capability) and digital literacy [20,21]. For citizens, spatial enablement is very close to Lévy's [55] notion of 'spatial capital', built on Bourdieu's notion of social capital. Spatial capital refers to the resources accumulated by stakeholders that allow them to take advantage of spatial dimensions of society [55]. To develop smarter cities, it is critical to develop and mobilize citizens' spatial capital and, thus, their social engagements.

This shows to what extent social inclusion is a major issue. A smart city must indeed build favorable conditions so that all citizens, including the disabled, can develop and mobilize their spatial

skills and thus find their place in the city. This is typically what this paper tackles. Through the mobilization and conjugation of the three aforementioned knowledge fields, i.e. the inclusive smart city approach, the DeafSpace and Disability Creation Process (DCP) model, as well as the concept of spatial skills, through two design fiction workshops, we intend to demonstrate paths for improving the spatial capability of hearing impaired users of cochlear implants.

## 2.4. Design Fiction

In this Special Issue's call for papers, universal design is defined as "*the design and composition of an environment so that it can be accessed, understood, and used to the greatest extent possible by all people regardless of their age, size, ability, or disability. An environment (or any building, product, or service in that environment) should be designed to meet the needs of all people who wish to use it*." Even if universal design has evolved a lot since its very first development after the Second World War [15], it is rather unlikely to develop "one size fits all" smart city solutions without reducing their usefulness. Of course, the main universal design principles (equity, flexible use, intuitive, easy use, perceptible information, tolerance to error, minimum physical effort, and dimension and free space for approach and use) [56] remain relevant in our own research.

Despite this, our position here, in particular considering the users targeted by this research, is that the inclusive nature of a smart city project essentially depends on taking into account certain specificities of deaf people, or rather the specificities of their interactions with urban spaces. We have then chosen to use design fiction. Design fiction is an evolving concept providing a powerful narrative-based tool to contextualize a future design and the needs, values, and experiences associated with it. It enables design collaborators to grasp prospective design ideas and more profoundly explore the implications of future technologies, practices and/or methods, something that becomes particularly useful when design mediates complex experiences for disabled people.

What is design fiction? According to the study conducted by Ahmadpour et al. [57], design fiction is characterized by a combination of six characteristics (taken from an extensive literature review) which allow users to move forward a different conceptual space:

- *"i. Design fiction suspends disbelief in change,*
- *ii. It often examines the implications presented by potential design in the context of technological conflict,*
- *iii. It enables discussion of social and political context,*
- *iv. It inspires discussion about desirable and preferable futures,*
- *v. It helps to reveal potential user concerns and uncertainties, and*
- *vi. It presents a disruptive space for emerging cultural artifacts."* [57]

Thanks to design fiction, future possibilities could be explored [58] using "speculative design" methods that draw on creative practice [59]. In our research, the rationale for using design fiction is two-pronged: first, it has allowed us to generate debate among participants in the two workshops [60,61], second, design fiction is a very interesting and efficient way of developing prototypes that help participants to collectively project themselves in a speculative future [58,59]. In addition, Nägele et al. [62] clearly demonstrate how design fiction constitutes a method and offers tools that are perfectly suited to users in vulnerable situations. Hales [63] considers design fiction as a "speculative turn" within contemporary design practice, characterized by its "multidimensionality". This capability of design fiction to create a "*discursive space within which new forms of cultural artefact (futures) might emerge*" [63] is another argument that justified the use of design fiction for this research, in which deaf culture has to be taken into account from the very beginning of the process.

## 3. Design Fiction Workshops

The participatory design fiction approach was therefore chosen as an operating mode, but also as a mode of user engagement, through the organization of two design fiction workshops [64]. Those two workshops were grounded in the principles of inclusive design. Users were engaged there as part

of a participatory and iterative process, based on the stimulation of their five senses [56]. The first workshop, "The Vibropod", was implemented during the first author's art, and research residency, carried out at the "Chambre Blanche", a self-managed artist center, from 27 February 2017 to 27 February 2018, while the second, "The Pointe-aux-Lièvres", was held in a Québec public park on 23 August 2018. Those two workshops offered participants exploratory sensory research activities grounded in a smart city context.

Reproducibility is still an issue in this kind of research method for various reasons. The first one is certainly due to the exploratory and inductive approach our workshops were essentially based on. Indeed, the research we have conducted has gone through a trial and error process. The Vibropod prototyping then evolved and was improved as we better understand the context, stakes and issues related to the engagement of people in our experimentations. Nevertheless, the following section provides details and information in order to better understand both the conceptual and technological sides of the Vibropod prototype, without compromising intellectual property-IP issues.

*3.1. The Vibropod*

The goal of the year-long residency at the "Chambre Blanche", was to create an accompanying object for deaf pedestrians. This year was dedicated to design, conceive and build the first Vibropod prototype, and ended by the fiction design evening/workshop.

The Vibropod is a sensory vibration device that supports urban spatiality (body geography and relationship with urban geography). In a way, it offers deaf people a sensory vibratory spatial grammar allowing them to interpret the sound signals of the urban environment according to their own skills (e.g., if a vibration is perceived in the right forearm then that means you should move to the right; if you feel a left-rear shoulder vibration, then you should look away to the left because there is potential danger). The idea is then to support deaf people in the use of their own sensory potential, to make their city interactions smarter and safer, by sensing urban vibrations (Figure 2).

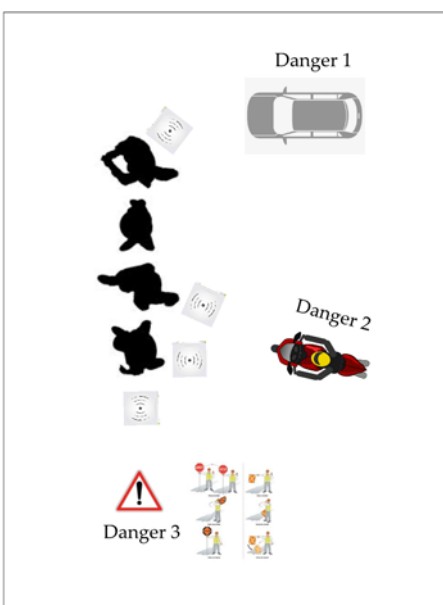

**Figure 2.** Sensing urban vibrations.

3.1.1. The Vibropod Design

Before becoming an object, the Vibropod was thought of as a space (room) composed of different interfaces which aim to accommodate one or more people (Figure 3). It was essentially a "machine" for translating sounds into vibrations. Then, as issues and parameters became clearer, and in order to put participants in context in a prospective use scenario, the focus turned to designing an identifiable

object. The Vibropod was thus conceived as a technological jewel. The design work followed two steps, based on the TRIZ theory (Russian acronym for Inventive Problem Solving Theory) [65,66]. Step 1 was spatialization, while step 2 was object design. The Vibropod prototype was designed (during the first part of the art and research residency) in a context where participants (at this stage essentially people working or collaborating with "The Chambre Blanche") were invited to participate in a series of design exercises (Figure 3).

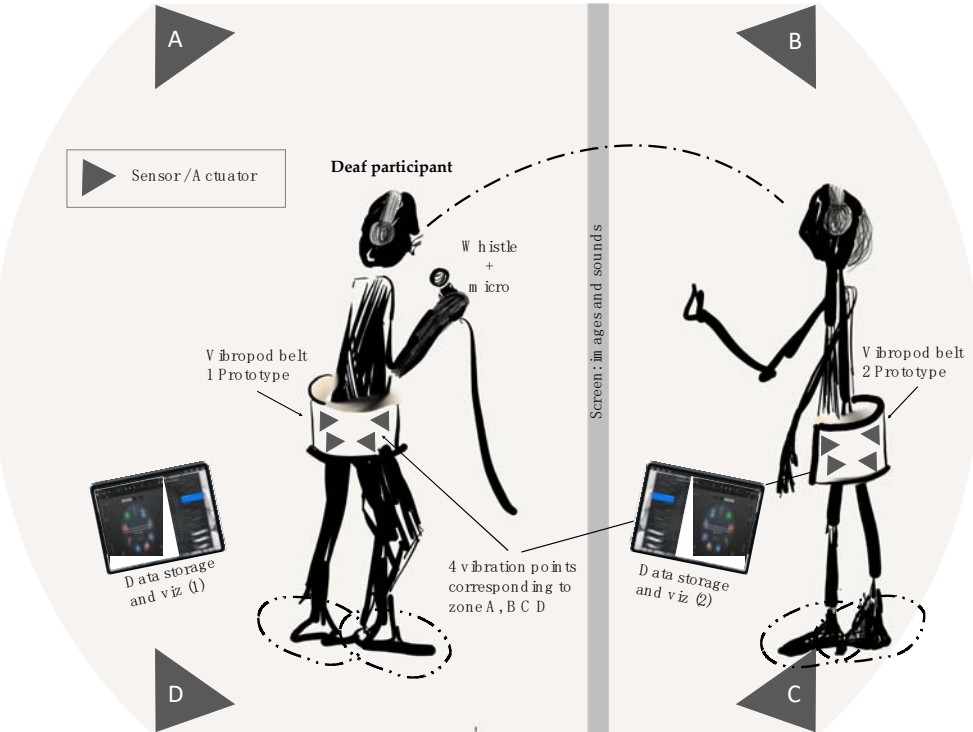

**Figure 3.** Vibropod design—spatialization step.

The TRIZ theory is a heuristic approach that aims to solve mainly technical innovation issues [67]. The TRIZ theory considers that technological progress generally follows a course that can be described by laws. In short, the problems encountered during the design of a new product have analogies with others, so analogous solutions be found and applied). Based on the analysis of a large number of patents [68], these laws suggest a procedure to be followed in order to innovate in terms of technology, in particular by exploring generic solutions, borrowed from other fields that have not yet been applied to the particular problem under study. The starting point of the TRIZ method is usually a question rather than a need (as in design fiction). The method suggests splitting the innovative design thinking process in 40 principles [64], which should be applied one after another, until the question is sufficiently answered.

The spatialization step (1) was set up as follows: A deaf person who navigates freely on a city street corner (scenario)/what sound information do they need to sense danger? (Question). Deaf participants were invited on stage (paired with people with or without hearing loss) in front of a microphone, to produce various sounds through different modes (whistle, footstep, speech, shout, etc.). Different kinds of sound interactions were tested: real whistle sounds, image sounds under construction, vibrating sounds from the Vibropod belt, and sounds felt by the effect of narrative repetition (Figure 3).

Those sounds were then translated into vibrations and returned to the participants. This vibratory experience was then shared in order to identify the vibrations, but above all, relevant sound/vibration pairs. During this development stage, vibrations took on a tangible character. The vibrations were analyzed following the TRIZ sequence up to the seventh principle.

- Segmentation (1): dividing the vibrations in independent as well as linked and scalable components, allowed us to proceed to a spatialization of body sounds;
- Extraction (2) and total quality (3): extracting only the relevant parts and properties of meaningful "types of sound/vibration/body part" associations;
- Asymmetry (4): replacing a symmetrical object by an asymmetrical one, as a way of locating participants in space (Figure 3);
- Combination (5): spatial and temporal integration of homogeneous or contiguous modes of sound production;
- Universality (6): designing a versatile device (in line with universal design), a device that can produce and sense different sound/vibration combinations;
- Nesting (7): designing an object which can fully contain another object, and could be fully contained by another one (Russian dolls).

The Design step (2): the rationale of the Vibropod was to help deaf people move safely in an urban environment by making potential dangers felt (sensed) through a vibratory modality (vibration is here considered as the materialization of sound in its simplest form). At this stage of the process, the Vibropod was transformed into a simplified and more accessible small device version.

On one hand, in terms of hardware, the device was not easily portable and wearable and installing it required space. On the other hand, the objective was to place the device in a real situation, which is to say in the context of use in urban mobility. Moreover, initially, a simplified version of the device (in a single nesting object) was developed (Figure 4), before being tested with participants (validation of the vibratory/sound information necessary to answer the initial question). This was the main focus of the design fiction evening.

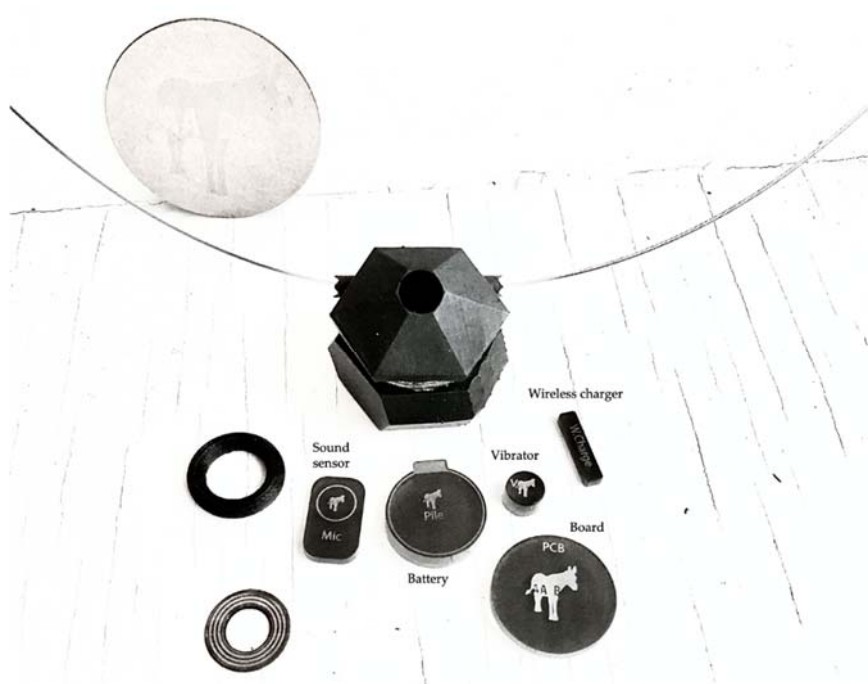

**Figure 4.** Vibropod components.

### 3.1.2. How Does the Vibropod Work on a Technical Level?

The outer shell of the Vibropod is the result of a three-dimensional (3D) printing process (Figure 4). It measures approximately 1 inch in diameter. The following main components are stacked on top of each other inside the shell (Figure 5):

- A circuit board control card (Simblee Bluetooth (BLE) Module—RFD77101: a small Arduino-programmable Bluetooth 4.0 (BLE) module allowing control of the IoT—Internet of Things) + RFduino—Simblee Starter Kit;
- A sound sensor (VCC 2.4-5.5 V Adjustable Gain GY-MAX4466 20-20 KHz Electret Microphone Amplifier MAX4466 Adjustable Amplifier Sensor Module);
- A vibrator (a VPM2 Vibrator: a Mini Motor generating a silent and intense vibration, in a metal case easy to put in place with the supplied adhesive);
- A Custom Rechargeable Lithium Power battery LIR2450 (3.6 v, 120 ± 5 mah, 0.2c ma, 1c ma, 24.5 mm × 5.0 mm, 5.3 ± 0.2 g);
- A Wireless Charging Module 5 V/300 mA (induction charger);
- A chain allowing the user to wear the device around the neck.

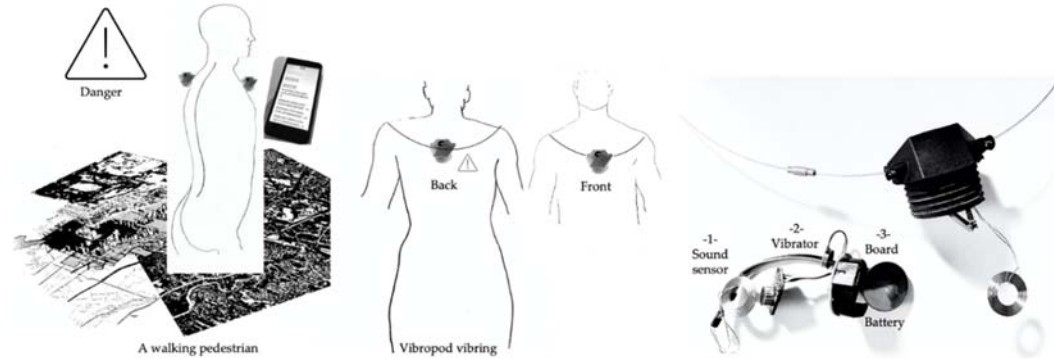

**Figure 5.** Vibropod working principles.

The operating principle of the Vibropod is quite simple. When an ambient sound (siren, horn, shout, etc.) is picked up by the microphone, it is transmitted to the vibrator via the control card. The vibrator then emits a vibration whose direction and intensity informs the user about the nature and origin of the danger. Additional configurable information (nature of the danger and approximate location of the source) is displayed on the user's smartphone through the app interface (Figure 5) connected to the Vibropod.

The stage of prototype development during the workshops involved limitations: the user had to wear two devices (Figure 5), one on the chest and the second on the back, so as to produce indications about the origin of a potential danger (front or rear). The current operational prototype is not yet equipped with the capacity to produce spatially oriented vibrations on the user's body corresponding to the geographical origin of the sound picked up by the sound sensor. In addition, the autonomy of this prototype is not compliant with a daily use. Energy consumption will have to be optimized and components will still have to be miniaturized.

3.1.3. The Vibropod Design Fiction Workshop

At the end of the residency year, a design fiction evening/workshop was organized with the aim of stimulating the senses usually needed to navigate city space and streets, and generating an emotional response from the participants in a sequential manner, as they are invited to step into the shoes of deaf pedestrians (Table 1). The workshop was structured around three steps (Figure 6):

- Reading an excerpt from "The concept of nature" by Whitehead;
- Followed by the movement of a red object in space (a billiard ball);
- Then, finally, viewing a music video.

**Table 1.** Design Fiction Workshop characteristics.

| Tasks | Participants | Duration | Recruitment Participants | Interaction Tools | Deliverables |
|---|---|---|---|---|---|
| **La Chambre Blanche (Artists Center)** | | | | | |
| Design, conception, production of the Vibropod | First author, a 3D printing technician, partners | One year | | | Vibropod prototype<br>- 3D printing<br>- choice and testing of materials and components |
| Design fiction evening workshop | 10 (general public, students) | Two hours<br>- 60 min presentations<br>- 60 min discussion | - Communication and call for participations in social media,<br>- Contacts list (center), | - Art performance,<br>- Face-to-face survey,<br>- Video projection, | - Survey questionnaire<br>- Discussion verbatim |
| **La Pointe-aux-Lièvres (urban place)** | | | | | |
| Design of the soundtrack to "dress up" the urban space | First author, music composer, filmmaker | One week | | | |
| Design fiction evening workshop | 10 (general public) | Two hours | - Communication and call for participations in social media<br>- Contact list (Quebec City and "Pépinière Espace Collectif") | - Sound experimentation,<br>- Discussion<br>- Capture of the event by a filmmaker | - Documentary film<br>- Verbatim of discussion |

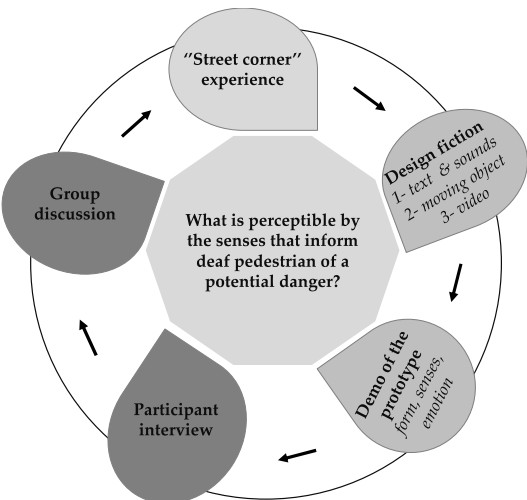

**Figure 6.** Protocol components.

Those three steps allowed participants, deaf or not, to project themselves into a situation where feelings of comfort and danger coexist. The main components of the Vibropod were presented (Figure 7) and a survey on the understanding of the mobility issues of deaf pedestrians was carried out.

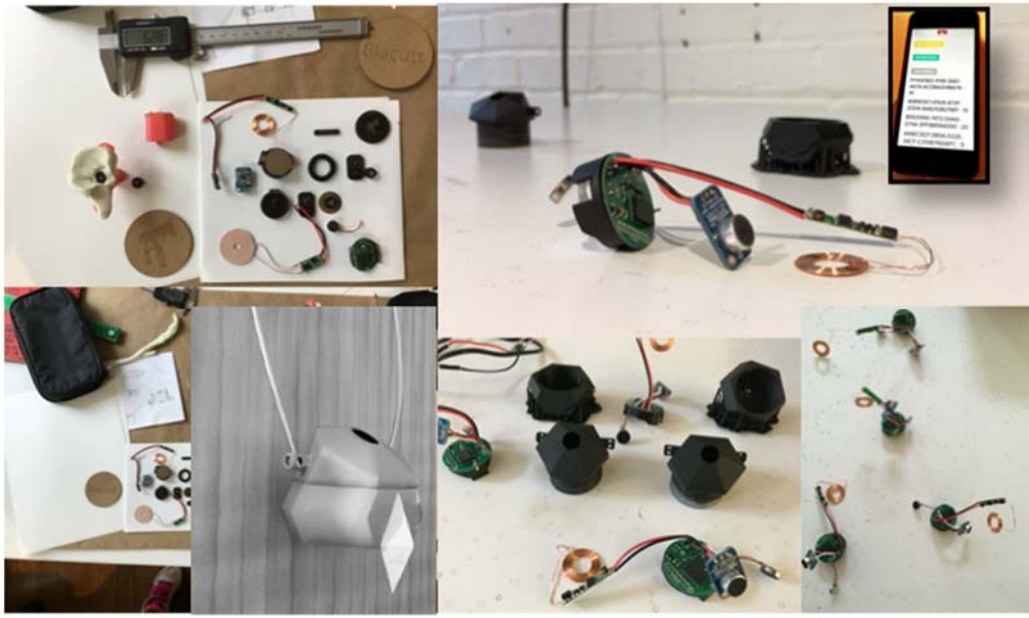

**Figure 7.** Sample of the Vibropod technological jewel (©A.-A. Blacutt).

*3.2. The "Pointe-aux-Lièvres" Urban Workshop*

The "Pointe-aux-Lièvres" urban workshop was the continuation of the "Vibropod" one. Indeed, the first workshop highlighted the importance for participants to be immersed in a "real urban context" (city) to give his experience a greater sensory range, and to have access to more explicit information (sounds/vibrations) and facts about deafness, in order to better understand its realities.

Those points helped guide the scenario for the design fiction workshop. Normal-hearing participants were invited to take part in a sensory experiment, bringing them into a simulated state of deafness, in the context of an ephemeral urban place located in a real Quebec urban place, where different modes of mobility (active and passive ones), and various uses and modes of occupation of urban space (sports, culture, etc.) coexist (Table 1).

This design fiction evening, called "Et si le son avait une crevaison" (and what if there was a hole punched in sound), was presented (and funded) in response to a call for tenders from the "Pépinière Espace Collectif", on the Pointe-aux-Lièvres site, an urban public park and place in Quebec City, on 23 August 2018. The activity was designed for a large audience of varying ages and cultures. The inclusive nature of the proposal was an invitation to act with the goal of producing reflections for social innovation in a smart and inclusive city context. A professional musician and a filmmaker joined the team.

Filmmaker Jean-Philippe Nadeau-Marcoux scripted his filming of the workshop as a documentary. His narrative was chronological, following actions in relation to the music, participants, and the environment. After providing context, the first author supervised the evening in order to guide the twenty participants.

Musician Martien Bélanger presented a musical performance (Figure 8) aimed at setting up a conversation on the realities of deaf pedestrians. The role of this musician was very important, as he offered a singular musical performance composed specifically for the event. The musical arrangement was performed live and in real time, in alignment with how the event was set to unfold.

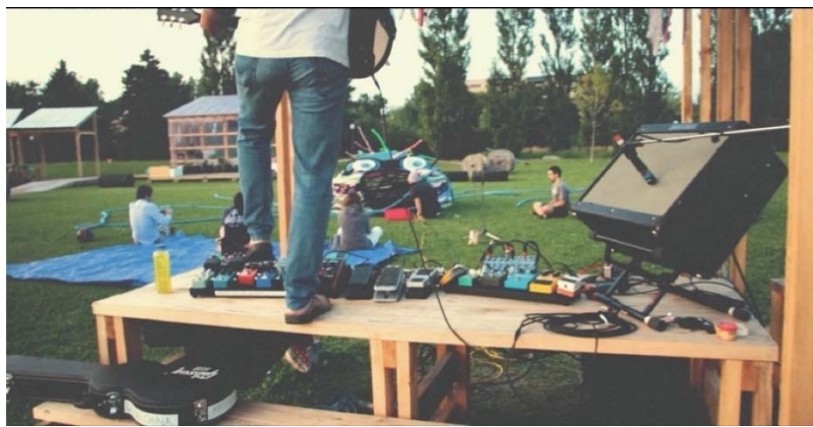

**Figure 8.** Workshop set-up (©A.-A. Blacutt).

At the opening of the workshop, participants were invited to listen to the original musical piece, which lasted approximately 10 min. This musical piece reflected the different aspects and intensities of sound and noises of an urban space. Afterwards, participants were gradually plunged into a state of deafness (ear plugs were provided). Finally, a group conversation took place, focusing on giving participants an opportunity to share their impressions and feelings. The goal of this two-hour workshop was to initiate a conversation on the issues of hearing loss in urban environments.

The workshop produced a soundtrack including the testimonies of the participants, as well as a short documentary film. This offered the possibility of analyzing the behaviors and reactions of people immersed in a situation of deafness.

## 4. Discussion

These two workshops were set in a logical sequence, with the second (Pointe-aux-Lièvres) building on conclusions drawn from the first (Vibropod), and transposing the experiment in an actual urban context, connected to a smart city vision. They constitute part of the operational framework of the first author's doctoral thesis. They made it possible to open up two main avenues of reflection and discussion, relative to the use of design thinking as a means of addressing the specific issues of deaf pedestrians in a smart city context.

### 4.1. Methodological Contributions

On the methodological side, design fiction and, from a more technical standpoint, TRIZ theory and methods, provided very relevant and useful tools to address smart city inclusivity issues, especially with regards to impaired citizens. Starting from the question, rather than from needs, made it possible to turn more classic points of view on their heads. As claimed by the DCP model, inclusivity issues of impaired people do not originate from the people themselves, but from an environment that is not compliant. Moreover, questioning the environment itself, rather than seeking to understand the specific needs of people with disabilities, is a way of considering a smart city project from the perspective of social innovation.

In the case study presented here, design fiction makes it possible to engage participants without barriers or preconceptions, in a singular mode of understanding (performance, staging and prospective projection) of the specific issues of interactions between deaf pedestrians and people with normal hearing. Indeed, the "handicap" situation is not unidirectional. It is the result of an inadequacy of the urban environment to the particularities of deaf people, but it is also a disturbance of individuals' capacity to communicate with each other.

For its part, the rich exchange on sound spatialization and objectivation created by design fiction provides a number of opportunities to open up reflections about disability in general and deafness in particular, not only for people with hearing impairments, but also for normal hearing

people. This makes it possible to sensitize and engage all people in a more inclusive manner. Moreover, this type of qualitative research method allows meetings and exchanges in which the practice of design is a vector of communication. Design fiction materializes real-life situations through fictional devices used to highlight participants sensory experience (vibrations, sounds, feelings of safety or insecurity). The highlighting of communication through design fiction becomes a mode of mediation. The devices are thus developed from what emerges when people come into direct contact with a problem/question that can potentially affect anyone.

Finally, these two workshops highlighted how interdisciplinary dialogue offers rich and unique opportunities to renew the field of reflection, methodological development, and innovation in inclusive smart city research.

### 4.2. Spatial-Enablement of Hearing Impaired People

Participants who took part in the different stages of the design fiction process (the performance component in particular) were very eloquent about the risk they live with on a daily basis in their urban journeys. The implicit but major role of spatial skills (some of which have been revisited from the DeafSpace markers standpoint) in the spatialization capacity of deaf people leads us to consider another way of including them in the city.

The event (the performance) in Whitehead's perspective [69] underscores the spatial and temporal aspects of deafness. It recreates what deafness is in nature and, thus, allows participants to take place in the space and time of an urban park, in a sort of sensory fiction. Of course, they do not become deaf through having plugs in their ears, but they approach the sensation of deafness via a musical performance that pushes forward, draws out, fades, and almost completely disappears. This is how design fiction acts on perception and works to open discussions through a concrete experience. Indeed, being able to access the sensory capacities and understanding of cognitive capacities (spatial skills in particular) required for daily urban mobility, in connection with those of deaf people, opens the door to a more inclusive consideration of what the deaf community calls Deaf Gain.

In terms of deaf people's spatial enablement, the experiment highlighted the usefulness of thinking about the spatialization of senses on a scalable perspective: from the body space to the geographical space. Indeed, the segmentation (TRIZ) of vibrations in different parts of the body was related to the spatialization of danger sources (Figure 2). The use of DeafSpace markers (Figure 1) combined with the principles of TRIZ theory (segmentation, extraction, combination, and nesting, in particular) offers the possibility of better understanding what types of spatialization allow a deaf person to feel the most secure in an urban environment.

Detecting a threat or feeling a risk, locating them, delimiting their spatial scope, finding a bypass, are all spatial skills activated during the design fiction workshops, not by sight and hearing, but rather by sight and paired sensing of vibrations. The vibratory spatialization is certainly an innovative way to consider deaf persons' urban inclusion. The use of the second prototype of the Vibropod (Figure 4), a kind of technological "Russian doll" is certainly capable of producing this vibratory spatialization in the field. This will be part of the work planned for the next couple of years, in the context of A.-A. B.'s Ph.D. thesis. The idea is to explore how the Vibropod could be enhanced, based on the second workshop's results, in order to move towards a Smart DeafSpace.

### 5. Conclusions

In conclusion, we would like to emphasize the concept of Smart DeafSpace and its relevance in making a smart city more inclusive, in particular for deaf people. Smart DeafSpace could be seen as an inclusive component of smart city projects. This does not necessarily mean that we have to radically change the physical urban design and planning practices, but rather that vibrations and other DeafSpace markers should be integrated in the digital urban layer from the very beginning of the project. This is in line with the idea of not considering deafness as a limit or a disability, but rather through a force or a complementary sensory capacity of deaf people.

Experiments still need to take place, in order to explore potential interactions between body, space and sound/vibration, by focusing attention to the exchange of sound information offered by the Vibropod. There is indeed a whole set of information inputs and outputs in relation to human beings and their mode of sound perception through the phenomenon of translation (sound/vibration) offered by the Vibropod device. The objective would also be to allow participants in urban design projects to test certain "body modifications" to highlight otherness and the different experiences of sound spatialization linked to the singularity of deaf people in particular. It would be interesting to introduce prototypes of this type of device inside future smart city projects.

This paper aimed to show to what extent increasing spatial-enablement, and having a better control of spatial skills provides deaf people with new skills, to improve the use of technology in support of urban mobility, as well as to provide greater safety in urban environments. Our work opens the door to various researches in the fields of smart cities and urban design. We conducted two design fiction workshops based on the use of the DeafSpace markers and the TRIZ theory principles. These two workshops have enabled us to demonstrate that considering the singularity of the human being as a real acoustic material constitutes an innovative opportunity to improve the role of universal design in smart city projects, as well as their inclusiveness.

**Author Contributions:** The work presented in this paper is part of the A.-A.B.'s doctoral research project. As an artist, entrepreneur and, Ph.D. student in design and social innovation, A.-A.B. has notably focused on providing conceptual frameworks dedicated to deafness issues and challenges, and managed the realization of the two workshops. S.R. is a full university professor in geospatial science, working on smart and inclusive city projects. S.R. has provided the smart city and spatial oriented contents, as well as written the first plan draft of the paper. Conceptualization, A.-A.B. and S.R.; methodology, A.-A.B.; validation, A.-A.B. and S.R.; formal analysis, A.-A.B. and S.R.; writing—original draft preparation, A.-A.B. and S.R.; writing—review and editing, A.-A.B. and S.R.; supervision, S.R. All authors have read and agreed to the published version of the manuscript.

**Funding:** This research was funded by the Chambre Blanche (Art and Science residency), the "Pépinière Espace Collectif", and the "Desjardins Caisse d'Économie Solidaire"—no Grant numbers available.

**Acknowledgments:** Authors acknowledge all of the volunteer participants of the two design fiction workshops.

**Conflicts of Interest:** The authors declare no conflict of interest. The funders had no role in the design of the study, in the collection, analyses, or interpretation of data, in the writing of the manuscript, or in the decision to publish the results.

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
