# Peer review of "When Design Fiction Meets Geospatial Sciences to Create a More Inclusive Smart City"

_smartcities, doi:10.3390/smartcities3040064_

Round 1
Reviewer 1 Report
The paper entitled "When Design Fiction meets Geospatial Sciences for a
3 more inclusive Smart City" is really interesting, and I think it is suitable for publication. I only have some suggestions for improving the paper.
The abstract should emphasize the originality and the focus of the research.
The introduction should be better structured, framing the paper in more detail. In relation to the concept of smart cities, I strongly suggest to authors to talk about the evolution from the era "smart cities 1.0" (based on technology) to "smart cities 2.0" one (following an human-centered approach). For these reasons, I strongly suggest to read and quote the following articles:
- Kamolov, S., & Kandalintseva, Y. (2020, January). The Study on the Readiness of Russian Municipalities for Implementation of the “Smart City” Concept. In Ecological-Socio-Economic Systems: Models of Competition and Cooperation (ESES 2019)(pp. 256-260). Atlantis Press.
- Garau, C., Desogus, G., & Zamperlin, P. (2020). Governing technology-based urbanism. The Routledge Companion to Smart Cities. (particularly the paragraph: From smart cities 1.0 to smart cities 2.0: what about the governance? pp. 159-162).
Authors must better organise the section "Discussions and Conclusions", by separating it, in order to facilitate the reading of the paper and also to concentrate the results of the research in the conclusions.
Author Response
Thanks for your useful and relevant suggestions and remarks, this is very much appreciated.
We have tried to do our best to answer your questions.
Please find the attached table for further details about what have been done.

Reviewer 2 Report
This paper has much to recommend it. It contains socially-important ideas. It could though be better organized and presented.
The first issue relates to the concept of ‘the smart city’ [not ’smart city’ as used throughout]. The authors circle around what they are trying to say, but it takes a while to land on their goal which emerges on Line 51: to improve “the characteristics of the environment rather than as the limits of the person”. That could be stated immediately to give the paper clarity.
A crucial second issue relates to deafness. At one or two points, deafness is conflated explicitly with cochlear implants. This is highly problematic due to the small numbers of the devices in use and the political issues that surround their use. More needs to be said on this and these issues confronted. I am also concerned a phrase such as “that Deaf people use” [141] is defining when the goal of the paper is inclusionary.
The paper also needs to tie its themes together. The front end is all about cities, but this is not mentioned again after the workshops. This needs to be resolved, especially given the journal in question.
The presentation is a little idiosyncratic and needs work. Some examples include:
As mentioned, the paper needs to use the term “The smart city”, or “a smart city”, not ‘smart city’
Avoid ending a sentence by running out of ideas […………] on several occasions.
While we still have apostrophes, please use them: ‘Cities' engineering’ , ‘Citizens’ engagement’……;
Some phrases are slightly un-colloquial: As regards as, As a matter of facts, ethics smart city platforms, Concrete experimentations.
Instead of writing ‘According to [53]’ the author should provide the name of the researcher being cited, in several places.
Avoid gendered language by using plurals: e.g. 233-240
Avoid the word ‘Properly’ in the context of behavior which seems directive or normative; use instead ‘appropriately’ or ‘effectively’
273 Despite [insert 'this'], our position here; 395 'dole jewel' [?];
“What if the sound had a flat” is ambiguous given the use of [sharps/flats] in music. English-English uses ‘puncture’ which might be better.
Author Response

(The authors gave the same response as above.)

Reviewer 3 Report
The article addresses an interesting topic for technological development to support people in an urban context and aims to make a relevant contribution to the social and inclusive dimension of Smart Cities. On the other hand, the combination of two dimensions not always of peaceful coexistence - design and technology - is welcomed.
However, we believe that the methodology used should be able to be reproduced by other researchers or other research projects. This idea ends up rendering the article very fragile.
Many aspects remain to be described both at the technological level - after all the Vibrapod is about?, how it works?, how it applies to the body?, what is its energy source?, what is its autonomy?, what limitations does it have?, ... - or at the level of detail of the workshops - one lasts one year and the other two hours? how many participants? How were they chosen? How did the workshop work?
The absence of these elements, the lack of precision and detail and the impossibility of reproducing the experience described here end up condemning the article to insignificance and even uselessness, despite the goodness of its objectives.
Section 3.1.1 is missing.
Author Response

(The authors gave the same response as above.)

Round 2
Reviewer 2 Report
I believe that the authors have undertaken a conscientious revision. BUT I notice that the large amount of new material also displays a large amount of incorrect English and simple typos. I point out some instances below but I encourage the authors to have the paper properly proofed as it is likely to be widely read and these all detract.
235 he medical ?
245 explantation ?
344-50 what intellectual property?
579 'Russian doll' is enough
Author Response
Suggested corrections have been done and the paper has been entirely revised by a native english spoken professional.
Reviewer 3 Report
It seems to me that the article is now in a condition to be published. I would only suggest a bibliographic reference to accompany/justify the idea of Smart City3.0 - Lines 50 and 51), like, for example:
Vishnivetskaya, A., & Alexandrova, E. (2019, March). “Smart city” concept. Implementation practice. In IOP Conference Series: Materials Science and Engineering, 497(1):12-19. IOP Publishing.
or,
Bednarska-Olejniczak, D., & Olejniczak, J. (2016). Participatory budget of Wrocław as an element of smart city 3.0 concept. Sborník příspěvků XIX. mezinárodní kolokvium o regionálních vědách Čejkovice, 15(17), 6.
Author Response
A Smart city 3.0 reference has been added line 51 as suggested.